# Urban birds' tolerance towards humans was largely unaffected by COVID-19 shutdown-induced variation in human presence

Peter Mikula [1,2,3,15] ✉, Martin Bulla [3,15] ✉, Daniel T. Blumstein [4], Yanina Benedetti[3], Kristina Floigl[3], Jukka Jokimäki [5], Marja-Liisa Kaisanlahti-Jokimäki[5], Gábor Markó[6], Federico Morelli[3,7], Anders Pape Møller[8,9], Anastasiia Siretckaia[3], Sára Szakony[10], Michael A. Weston [11], Farah Abou Zeid[3], Piotr Tryjanowski[1,2,12,16] & Tomáš Albrecht [13,14,16] ✉

The coronavirus disease 2019 (COVID-19) pandemic and respective shutdowns dramatically altered human activities, potentially changing human pressures on urban-dwelling animals. Here, we use such COVID-19-induced variation in human presence to evaluate, across multiple temporal scales, how urban birds from five countries changed their tolerance towards humans, measured as escape distance. We collected 6369 escape responses for 147 species and found that human numbers in parks at a given hour, day, week or year (before and during shutdowns) had a little effect on birds' escape distances. All effects centered around zero, except for the actual human numbers during escape trial (hourly scale) that correlated negatively, albeit weakly, with escape distance. The results were similar across countries and most species. Our results highlight the resilience of birds to changes in human numbers on multiple temporal scales, the complexities of linking animal fear responses to human behavior, and the challenge of quantifying both simultaneously in situ.

The actions taken to control the coronavirus disease 2019 (hereafter COVID-19) pandemic locked inhabitants in their dwellings and thus changed the pattern of human outdoor activities[1–9]. This situation created a quasi-experiment that offered a unique opportunity to study how rapid changes in human behavior affect wildlife[10–14]. The shifts in human presence and activity associated with the COVID-19 pandemic might have been particularly noticeable in urban areas[12,15]. Consequently, the COVID-19

shutdowns might have elicited complex effects on urban nature[10]. For example, some animals increased their abundances in human-dominated landscapes, moved to new areas, or shifted the timing and character of their activities; some of these effects were area- or species-specific[10,11,16–23].

To persist in urban habitats, wild animals must increase their tolerance to humans and human disturbance[24,25]. Indeed, on a gradient from natural to urban environments animals seem more tolerant of humans,

¹TUM School of Life Sciences, Ecoclimatology, Technical University of Munich, 85354 Freising, Germany. ²Institute for Advanced Study, Technical University of Munich, 85748 Garching, Germany. ³Faculty of Environmental Sciences, Czech University of Life Sciences Prague, Kamýcká 129, 16500 Prague, Czechia. ⁴Department of Ecology and Evolutionary Biology, University of California, 621 Young Drive, South, Los Angeles, CA 90095, USA. ⁵Arctic Centre, University of Lapland, PO Box 122, 96101 Rovaniemi, Finland. ⁶Department of Plant Pathology, Institute of Plant Protection, Hungarian University of Agriculture and Life Sciences, Ménesi út 44, 1118 Budapest, Hungary. ⁷Institute of Biological Sciences, University of Zielona Góra, Prof. Z. Szafrana St. 1, 65516 Zielona Góra, Poland. ⁸Ecologie Systématique Evolution, Université Paris-Sud, CNRS, AgroParisTech, Université Paris-Saclay, 91405 Orsay Cedex Paris, France. ⁹Ministry of Education Key Laboratory for Biodiversity Sciences and Ecological Engineering, College of Life Sciences, Beijing Normal University, 100875 Beijing, China. ¹⁰Department of Ecology, Institute of Biology, University of Veterinary Medicine Budapest, Rottenbiller u. 50., 1077 Budapest, Hungary. ¹¹Deakin Marine, School of Life and Environmental Sciences, Deakin University, Burwood Campus, 221 Burwood Highway, VIC 3125 Burwood Melbourne, Australia. ¹²Institute of Zoology, Poznań University of Life Sciences, Wojska Polskiego 71C, 60625 Poznań, Poland. ¹³Institute of Vertebrate Biology, Czech Academy of Sciences, Květná 8, 60365 Brno, Czech Republic. ¹⁴Department of Zoology, Faculty of Science, Charles University, Viničná 7, 12844 Prague, Czech Republic. ¹⁵These authors contributed equally: Peter Mikula, Martin Bulla. ¹⁶These authors jointly supervised this work: Piotr Tryjanowski, Tomáš Albrecht. ✉e-mail: petomikula158@gmail.com; bulla.mar@gmail.com; albrecht@ivb.cz

where human densities are high and human–wildlife interactions are generally harmless[24,26–29]. Additionally, within cities animal tolerance to humans seems to increase with increasing level of human presence (hereafter "human presence" or "human levels") but the association is often weak[30–33]. Whether the COVID-19 shutdowns induced changes in human activities and subsequently altered the urban landscape of fear and hence animal responses to human presence is still poorly understood[22,34]. However, if COVID-19 measures increased variation in human presence outdoors, this can be used to answer outstanding questions about how animals change their tolerance toward humans across different temporal scales.

Here, we explored how the increased variation in human presence outdoors, due to the COVID-19 shutdowns, altered avian tolerance towards humans across parks of five cities in five countries (Rovaniemi – Finland–, Poznań – Poland, Prague – Czechia, Budapest – Hungary, Melbourne – Australia). During the breeding seasons before the COVID-19 shutdowns (in years 2014, 2018, and 2019) and during the COVID-19 shutdowns (2020–2021), we measured avian tolerance towards humans by quantifying flight initiation distance, which is the distance from an approaching human at which a bird escapes[26,28,35–38]. Because the COVID-19 interventions increased the variation in human presence (i.e. number of humans) outdoors[39–41] and the temporal scale at which birds evaluate the changes in the landscape of fear and adjust their response is unknown, we tested how changes in human presence across multiple temporal scales (hour-to-hour, day-to-day, week-to-week and year-to-year) were associated with avian escape distance. We used (1) actual human numbers recorded during escape distance observation as a measure of hourly changes in human presence (hereafter "Number of humans"), (2) Google Mobility Reports data (hereafter "Google Mobility", https://www.google.com/covid19/mobility) as a measure of daily changes in human presence in parks, (3) the stringency of governmental restrictions (hereafter "Stringency index"; https://ourworldindata.org/covid-stringency-index; based on data originally published in[42]) as a proxy for weekly changes in human presence, and (4) years before COVID-19 shutdowns and years during COVID-19 shutdowns as a coarse proxy for yearly changes in humans presence (hereafter "Period"), which we validated with the other three, above described proxies of human presence. Although the baseline escape distance

(the intercept) and the strength of association (the slope) between flight initiation distance and human presence may vary across species, we expected that if individuals, regardless of species, responded to shutdowns, then this would induce a similar response. In other words, we expected a negative relationship between escape distance of birds and human presence. However, if shutdowns affected human presence in a country-specific manner, we would expect country-specific effects of shutdowns on avian tolerance. Moreover, if shutdowns had little effect on human behavior, we expected no change in avian tolerance.

## Results

We found no major and clear differences in the overall avian tolerance towards approaching humans on all temporal scales (Fig. 1). Specifically, (1) according to the expectation, the actual number of humans in the area when an escape trial was conducted (hourly temporal scale) correlated negatively, albeit weakly, with escape distance. (2) The relative number of humans within each day as measured by Google Mobility (daily temporal scale) and (3) weekly changes in number of humans as proxied by Stringency index (weekly temporal scale) were unrelated to avian escape distance. Finally, (4) avian escape distances were similar in the years before and during COVID-19 shutdowns.

The results were similar across countries (Fig. 1; supporting Table S1[43]), and robust to the presence and absence of the control for starting distance (i.e. the distance between a bird and an observer when the escape distance trial started; Fig. 1), which is a known correlate of escape distance[28,32,33,44], the changes in random effects structure of the models, or, in the case of Period, restriction of the analyses only to species sampled during both periods (Fig. S1, Table S2[43]). Moreover, the effect sizes from the global model, where all data were analyzed together, were similar to the meta-analytical effect sizes based on country-specific estimates (Fig. 1).

Altogether, while the response of humans to shutdowns was country-specific and the presence of humans in parks decreased in some countries, increased in others, or remained unchanged[7,8,45], avian tolerance, as we measured it, did not reflect such major changes (Fig. 1). The country-specific effect sizes are small and mostly center around zero (Fig. 1). The effects were also inconsistent within countries and sites as well as within species (Figs. 2–5 and S2–5[43]).

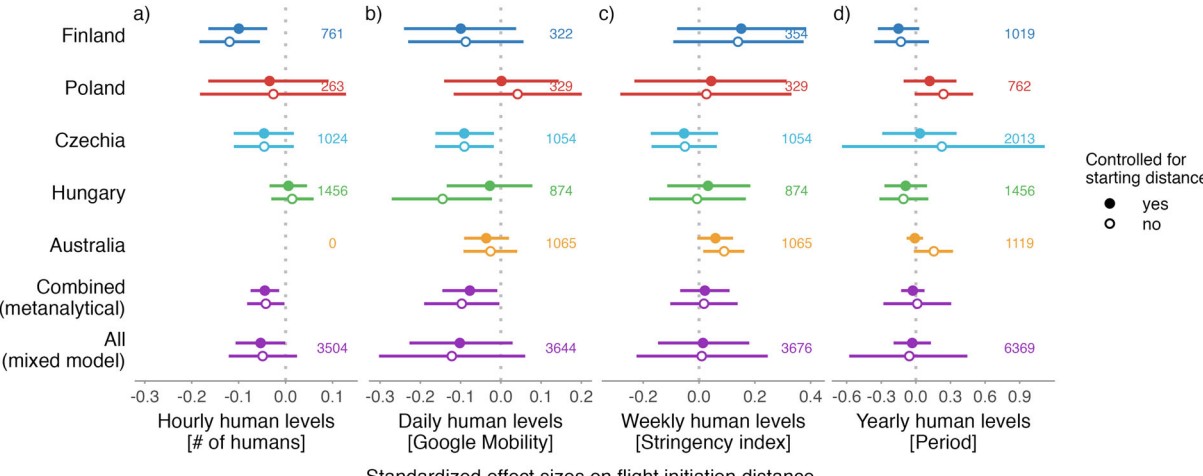

**Fig. 1 | Avian tolerance towards humans across four temporal scales. a–d** Avian tolerance according to (**a**) human levels (i.e. presence) during the escape distance trial (hourly scale) measured as number of humans within a 50-meter radius, **b** the day of the escape trial as proxied by Google Mobility (daily scale), **c** week of the escape trial as proxied by the stringency of governmental measures (weekly scale), and (**d**) the period of the escape trial, i.e. before vs during the COVID-19 shutdowns (yearly scale). The dots with horizontal lines represent estimated standardized effect size and their 95% confidence intervals, the numbers represent sample sizes. For the countries (i.e. country-specific models) and "All" (i.e. global models containing all

countries), the effect sizes and 95% confidence intervals come from the joint posterior distribution of 5000 simulated values generated by the sim function from the arm package[90] using the mixed model outputs controlled for starting distance of the observer (filled circles) or not (empty circles; Table S1[43]; for model specification details see Methods). For the "Combined", the estimates and 95% confidence intervals represent the meta-analytical means based on the country-specific estimates and their standard deviations (from the country-specific models), and sample size per country. Note that the effect sizes are small, and estimates tend to center around zero. For R-code generating the figure see interactive supporting material[43].

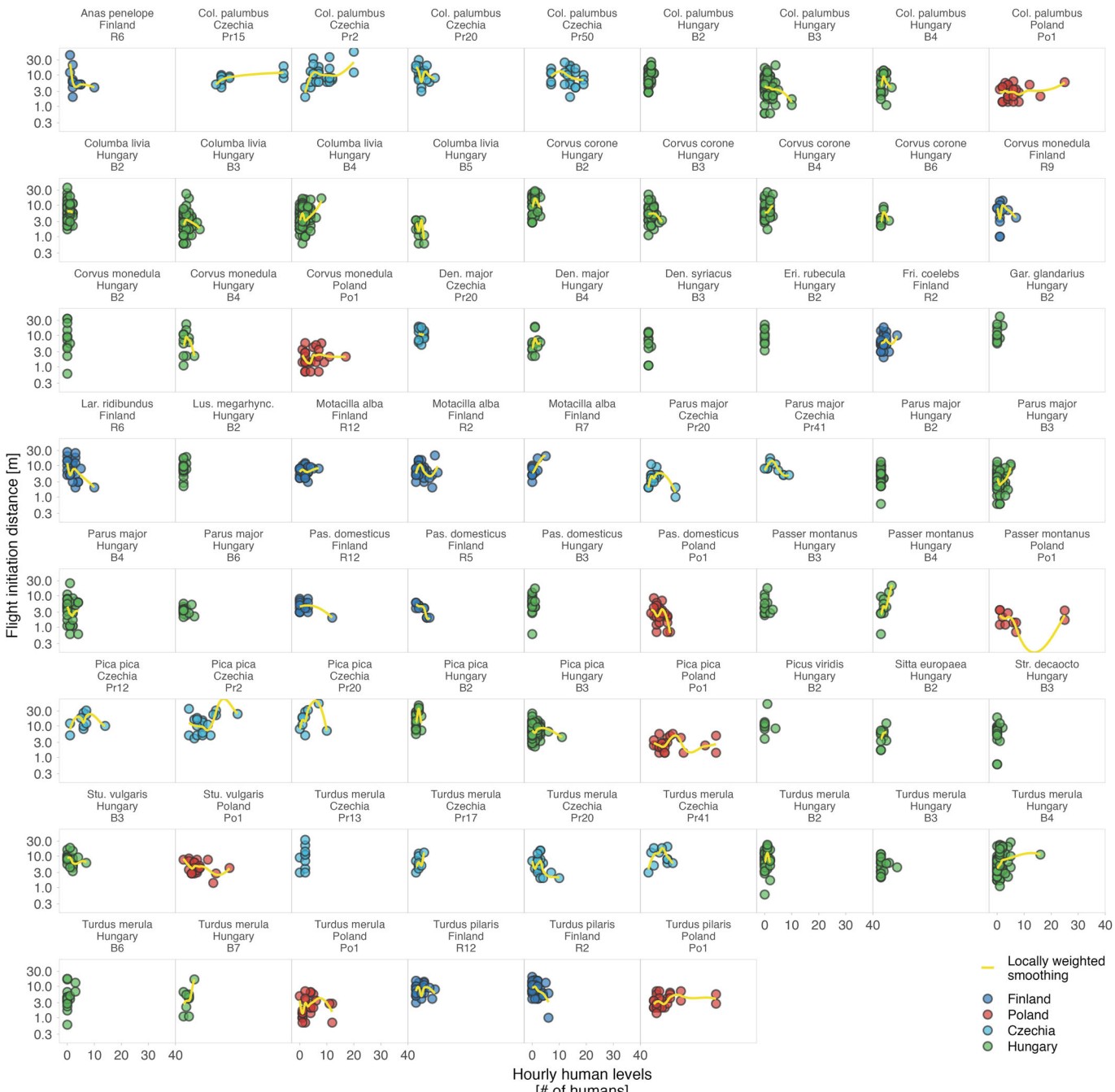

**Fig. 2 | Variation in avian tolerance toward hourly human numbers during the escape trials across species and sites.** Dots represent single escape distance observations of species at specific sites (e.g. park or cemetery) during the escape trial and not corrected for other factors such as starting distance of the observer. Dot colour highlights the country. Yellow lines represent locally weighted smoothing, a non-parametric local regression fitted with the ggplot function of the ggplot2 package[99] highlighting heterogenous (and usually unclear—close to zero) within- and between- species trends. Some species lack trend lines because data distribution hindered the smoothing and visualised are only data for species-site combinations with ≥10 escape distance observations and where number (#) of humans was estimated. The y-axes are on the log-scale. Panels are ordered alphabetically according to species names, then country and site identifier. Abbreviated genus names represent Col Columba, Den Dendrocopos, Eri Erithacus, Fri Fringilla, Gar Garrulus, Lar Larus, Lus Luscinia, Pas Passer, Str Streptopelia, Stu Sturnus, and abbreviated species name megarhync megarhynchos. For R-code generating the figure see interactive supporting material[43].

When exploring the patterns of human presence, we found that the actual number of humans at the time of an escape trial correlated weakly, and in a country-specific manner, with relative daily human presence (Google Mobility; Fig. 6) and weekly human presence (Stringency index; Fig. 6). As expected, (i) daily human presence generally correlated negatively, albeit weakly, with weekly human presence (Fig. 7), and (ii) relative daily human presence was generally lower during years with COVID-19 shutdowns (2020, 2021) than in 2022, a post-shutdown year

(Fig. 8). However, day-to-day variation in human presence seems larger (Fig. 8, middle plots) than changes induced by shutdowns (Fig. 8, left and right plots). In other words, the COVID-19 shutdown's influence on between-year differences in human numbers might have been negligible in studied countries. Yet, we cannot also exclude a possibility that the shutdowns changed human activities; hence, our year-to-year findings may reflect the resilience of the birds to changes in human activities.

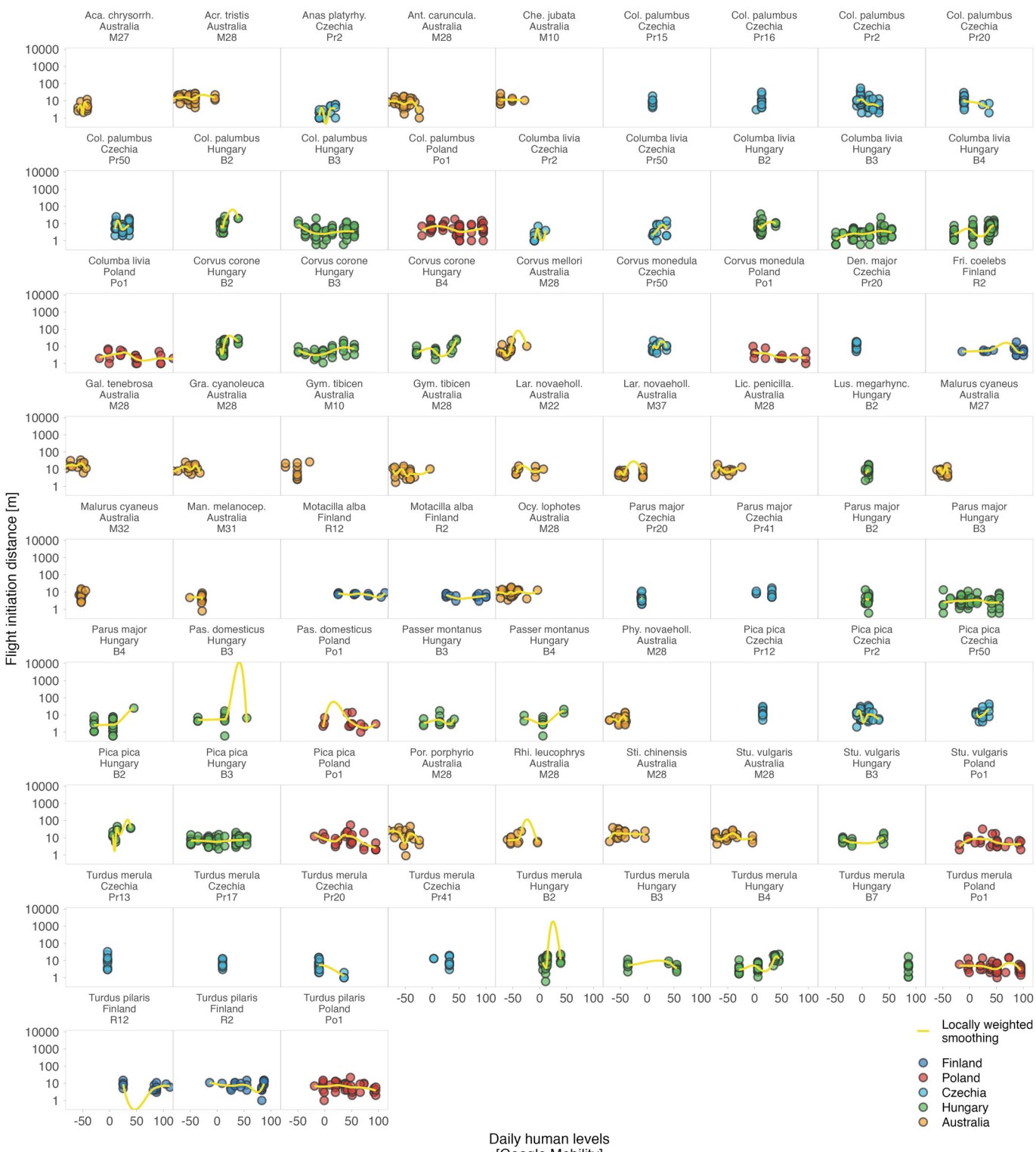

**Fig. 3 | Variation in avian tolerance toward daily human levels (Google Mobility) across species and sites.** Dots represent single escape distance observations of species at specific sites (e.g. park or cemetery) and not corrected for other factors such as starting distance of the observer. Dot colour highlights the country. Yellow lines represent locally weighted smoothing, a non-parametric local regression fitted with the ggplot function of the ggplot2 package[99], highlighting heterogenous (and usually unclear–close to zero) within- and between- species trends. Some species lack trend lines because data distribution hindered the smoothing and visualised are only data for species-site combinations with ≥10 escape distance observations, for which Google Mobility data were available. The y-axes are on the log-scale. Panels are ordered alphabetically according to species names, then country and site identifier. Abbreviations in the species names represent Aca. chrysorrh. Acanthiza chrysorrhoa, Acr Acridotheres, Anas platyrhy. Anas platyrhynchos, Ant. caruncula. Anthochaera caruncula, Che Chenonetta, Col Columba, Den Dendrocopos, Fri Fringilla, Gal Gallinula, Gra Grallina, Gym Gymnorhina, Lar. novaehol. Larus novaehollandiae, Lic. penicilla. Lichenostomus penicillatus, Lus. megarhync. Luscinia megarhynchos, Man. melanocep. Manorina melanocephala, Ocy Ocyphaps, Pas Passer, Phy. novaeholl. Phylidonyris novaehollandiae, Por Porphyrio, Rhi Rhipidura, Sti Stigmatopelia, and Stu Sturnus. For R-code generating the figure see interactive supporting material[43].

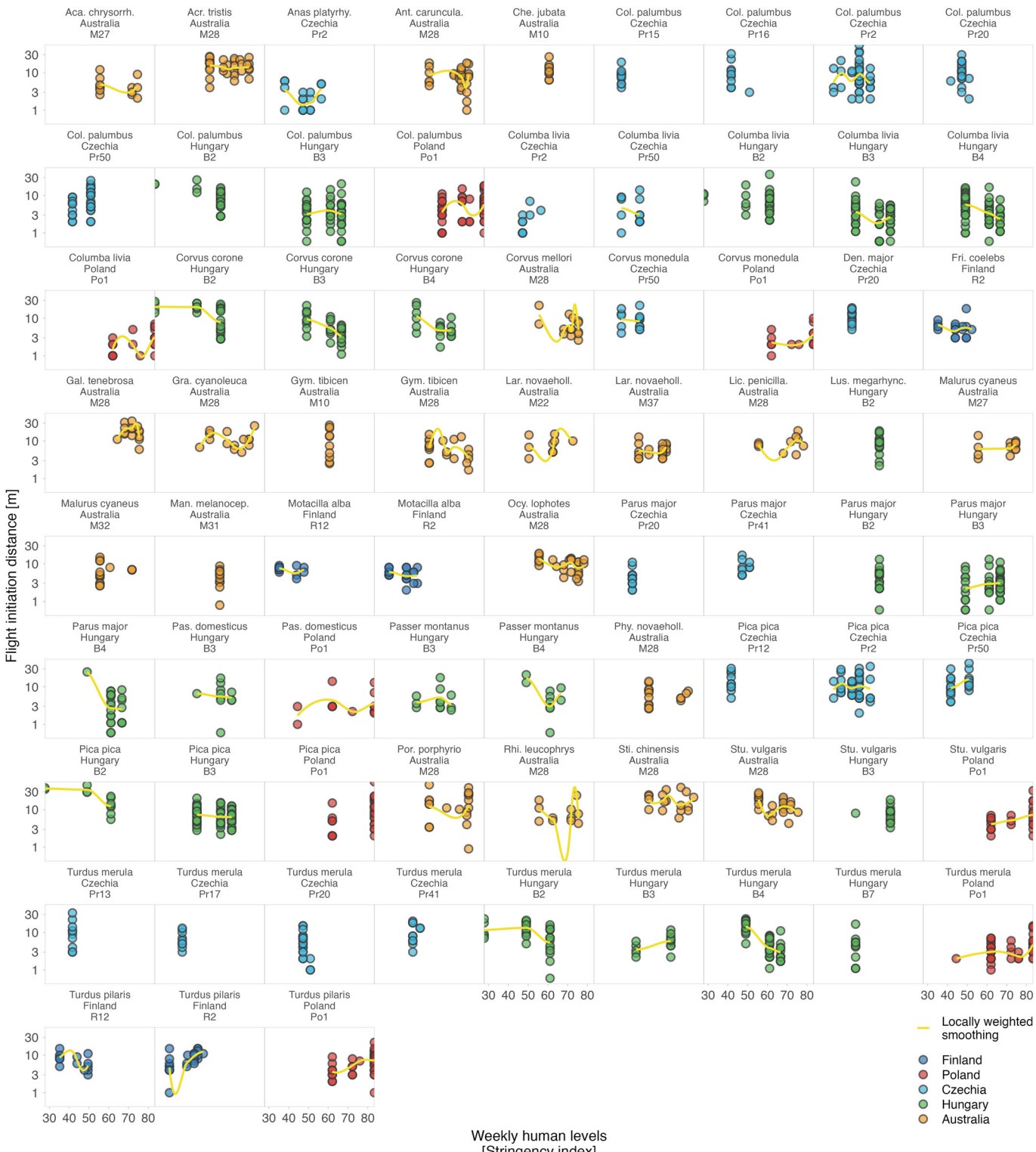

**Fig. 4 | Variation in avian tolerance toward weekly human levels (proxied by Stringency index) across species and sites.** Dots represent single escape distance observations of species at specific sites (e.g. park or cemetery) and not corrected for other factors such as starting distance of the observer. Dot colour highlights the country. Yellow lines represent locally weighted smoothing, a non-parametric local regression fitted with the ggplot function of the ggplot2 package[99], highlighting heterogenous (and usually unclear – close to zero) within- and between- species trends. Some species lack trend lines because data distribution hindered the smoothing and visualised are only data for species-site combinations with ≥10 escape distance observations, for which Stringency index data were available. The y-axes are on the log-scale. Panels are ordered alphabetically according to species names, then country and site identifier. Abbreviationed species names represent Aca. chrysorrh. Acanthiza chrysorrhoa, Acr Acridotheres, Anas platyrhy. Anas platyrhynchos, Ant. caruncula. Anthochaera carunculata, Che Chenonetta, Col Columba, Den Dendrocopos, Fri Fringilla, Gal Gallinula, Gra Grallina, Gym Gymnorhina, Lar. novaehol. Larus novaehollandiae, Lic. penicilla. Lichenostomus penicillatus, Lus. megarhync. Luscinia megarhynchos, Man. melanocep. Manorina melanocephala, Ocy Ocyphaps, Pas Passer, Phy. novaehol. Phylidonyris novaehollandiae, Por Porphyrio, Rhi Rhipidura, Sti Stigmatopelia, and Stu Sturnus. For R-code generating the figure see interactive supporting material[43].

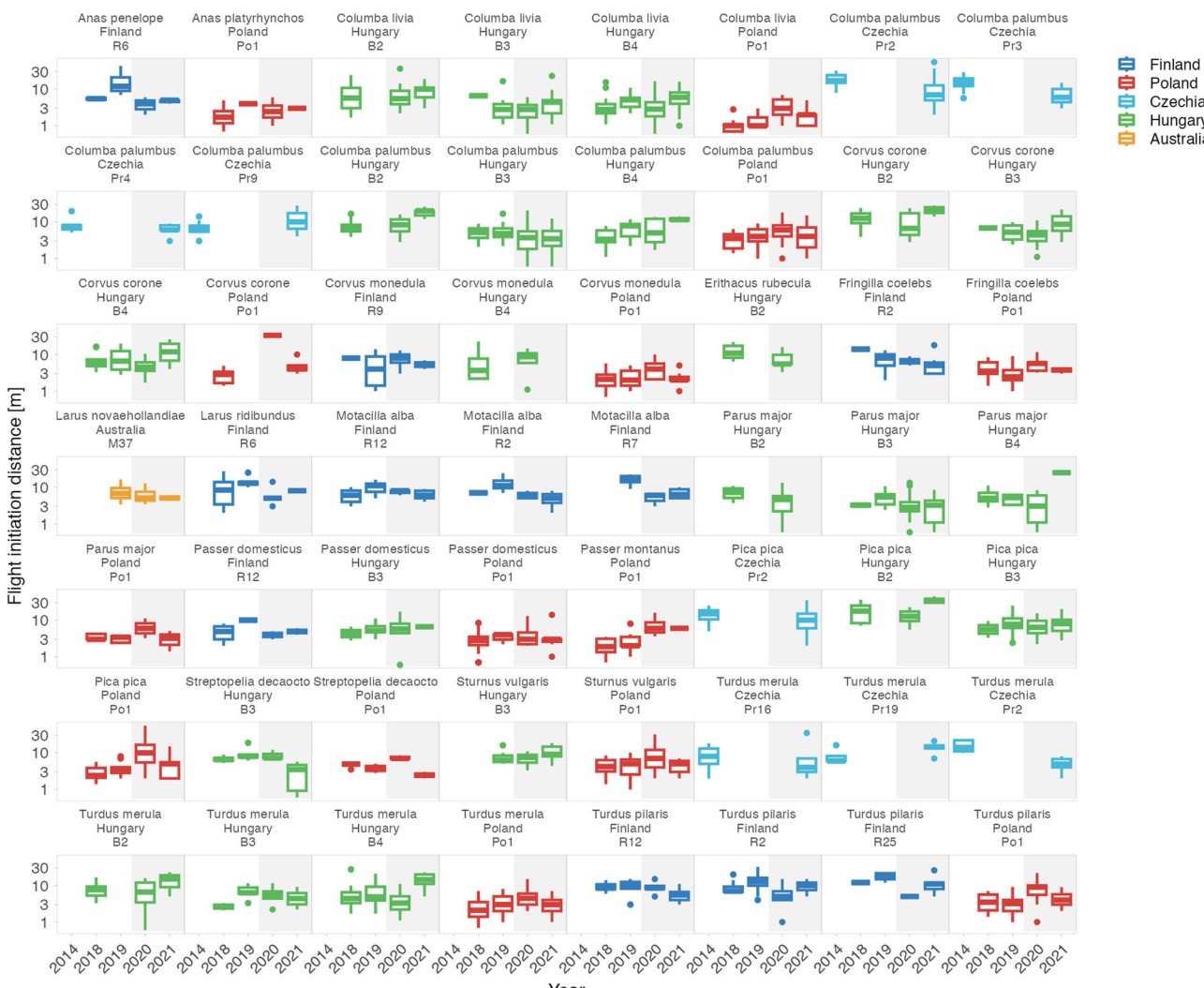

**Fig. 5 | Between-year variation in avian tolerance toward humans across species and sites.** Panels are ordered alphabetically according to species names, then country and site identifier within each country (e.g. specific park or cemetery). Boxplots outline colour highlights country, background colour indicates Period (white: before the COVID-19 shutdowns; grey: during the COVID-19 shutdowns). Boxplots depict median (horizontal line inside the box), the 25th and 75th percentiles (box) ± 1.5 times the interquartile range or the minimum/maximum value, whichever is smaller (bars), and the outliers (dots). Included are only species–site combinations with ≥5 observations per Period. The y-axes are on the log-scale. Note the lack of consistent shutdowns effects within and between species, sites and countries. For R-code generating the figure see interactive supporting material[43].

## Discussion

We used flight initiation distance as a measure of avian tolerance toward humans and capitalized on COVID-19 shutdowns that induced variation in human presence in parks to reveal that, in general, urban bird populations exhibited no major shifts in their tolerance toward humans in response to fluctuations in human presence across different temporal scales, i.e. hourly (actual number of humans during the experimental trial), daily (proxied by Google Mobility), weekly (proxied by Stringency index), and yearly (proxied by whether observation was conducted in years before or during COVID-19 shutdowns). These findings were largely consistent across countries, and independent of model specification. The trends for avian tolerance towards humans to increase with the increasing hourly (and to lesser extent also daily) human presence outdoors were small and weak.

The weak trends for birds to increase their tolerance to humans (i.e. tolerate closer approaches by humans) when human presence increased within short temporal scales, as well as the species- and area-specific nature of this effect that we report, are in line with previous studies[32,33,44,46,47]. Altogether, these results indicate that, in some situations, urban birds may be able to flexibly adjust their tolerance to short-term changes in human presence. Behavioral flexibility plays a role in short-term adjustments[31,47,48], although our

large-scale and multi-taxonomic comparison indicates that the room for behavioral adjustments may be limited for urban birds (e.g. because environmental filtering has left urban areas with mainly bolder individuals; see below)[40,45].

Our across-species findings that urban birds do not seem to react to the change in human presence on longer temporal scales, such as between weeks or years, is novel as we are not aware of a single study, which directly tests for such effects. Rather, previous studies investigated whether animal tolerance has changed since urbanization, regardless of actual levels of human presence[49,50]. However, a recent study using individually marked dark-eyed juncos (*Junco hyemalis*) showed, similar to our findings, weak effects of the COVID-19 shutdowns on flight initiation distance in urban birds[22]. The weak response of birds to long-term changes in human presence may indicate that (i) birds react to long-term shifts in human behavior in a nonlinear manner (which has to our knowledge never been tested) and/or (ii) such shifts in human behavior have to reach a threshold value to trigger observable changes in bird behavior. Thus, the changes in human behavior induced by the COVID-19 shutdowns, might have been insufficient to reach such threshold, since human presence in parks greatly fluctuates over the day, across weeks and seasonally (Fig. 6; e.g.[7,39–41]), perhaps explaining why

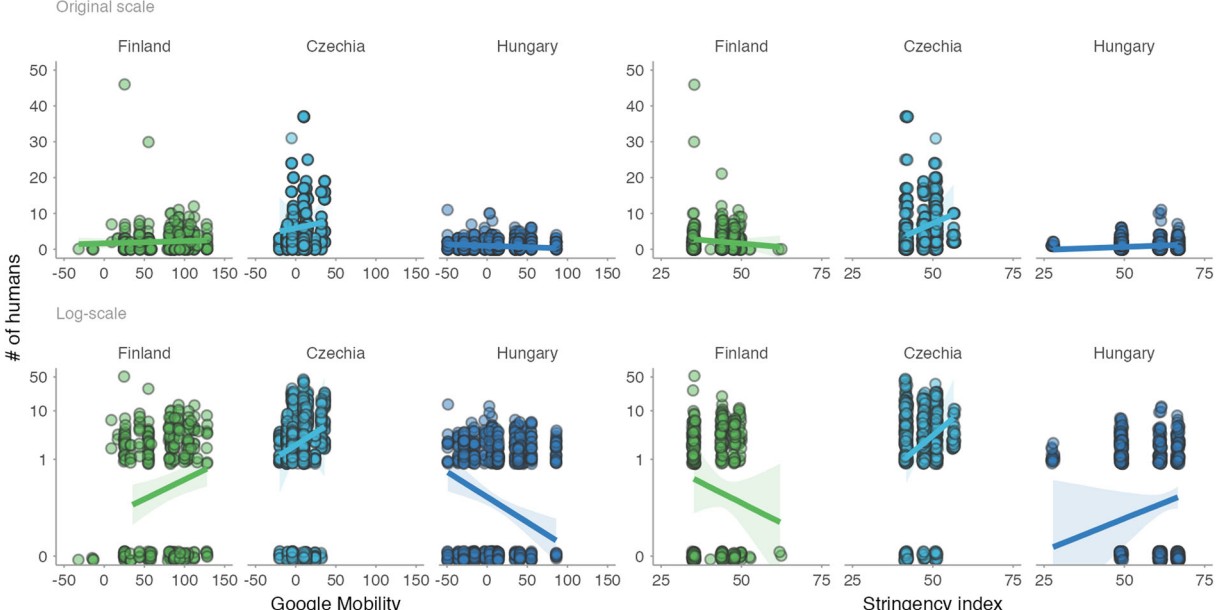

**Fig. 6 | Numbers of humans within 50 meters during the escape trial in association with daily human levels in parks (Google Mobility) and stringency of antipandemic governmental restrictions (Stringency index).** Dots represent individual data points (on original or log-scale), jittered to increase visibility. Lines with shaded areas represent predictions with 95% CIs from mixed effect models that controlled for the year (in case of Finland and Hungary) and non-independence of data points by including day of the week within the year as a random intercept and Google Mobility or Stringency index as a random slope (Table S3a–d[43]). Human numbers during the escape trials were missing for Australia (all years) and Poland (during shutdowns). Note the weak and country-specific associations. For R-code generating the figure see interactive supporting material[43].

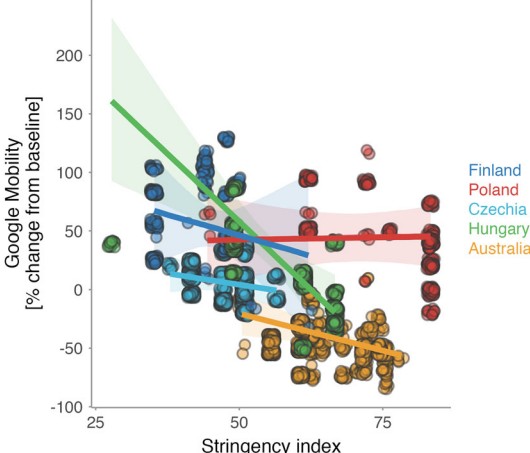

**Fig. 7 | Association between daily human levels in parks (Google Mobility) and the stringency of antipandemic governmental restrictions (Stringency index).** Lines with shaded areas represent predicted relationships from country-specific mixed effect models controlled for the year and non-independence of data points by including weekday within the year as a random intercept and Stringency index as a random slope (Table S3e[43]). Dots represent raw data, jittered to increase visibility, for days within which we collected escape distances in each city. Color indicates country. Note the generally negative but weak association between Google Mobility and Stringency index. For R-code generating the figure see interactive supporting material[43].

the occurrence of wildlife in the cities did not always increase during the COVID-19 shutdowns[17,19]. Whether similar shifts in human presence would induce changes, for example, in escape distances of rural birds that are not tolerant of humans and tend to have longer escape distances[26,28], awaits testing. Moreover, urban areas other than parks and cemeteries might have experienced stronger changes in the human presence (perhaps reaching a

required threshold), but whether birds changed their escape response there is also unknown.

To test whether the response of birds to changing levels of human presence is non-linear or based on threshold levels of human presence, repeated escape distance trials of marked individuals during various times of day and different days of the week, as well as under various natural or experimentally induced levels of human presence and activity are needed. However, most studies that investigated escape responses of animals towards humans and used marked individuals did not expose these individuals to the various levels of human disturbance ([48,51–53]; but see[22]).

Our findings also highlight the socio-ecological complexities of linking wildlife and human behaviors. In other words, the within-day and day-to-day fluctuations in human presence and outdoor activities raise questions of how and when to measure human presence and activity in field studies on animal tolerance towards humans. For instance, we still do not understand whether animals respond to the overall daily human presence and activity, or whether they flexibly adjust their behavior to levels of human presence and activity at a given time. We also acknowledge the substantial evidence that birds can discriminate between different forms of human activity, meaning that metrics of human abundance may not directly reflect birds' risk perception[54–56]. Finally, we acknowledge that some of our data were not designed to answer questions about short-term (mainly hourly) adjustments. Thus, it is possible that despite high human densities at some days of the week or during specific times of the day, animals may shift their activities accordingly (e.g. forage when human activity is low or rest and hide when human presence is high). To better understand this phenomenon, again repeated sampling of marked individuals within and across days is needed.

The general lack of response to the changes in human presence we described, has three possible implications. (i) Most urban bird species in our sample may be relatively inflexible in their escape responses because the species may be already adapted to human presence. In other words, natural selection on escape behavior of sampled urban birds might be at an optimum, and departing from this may be difficult. Moreover, bird species that colonized urbanized areas relatively earlier are more tolerant toward humans and also their escape responses to human approaches are less

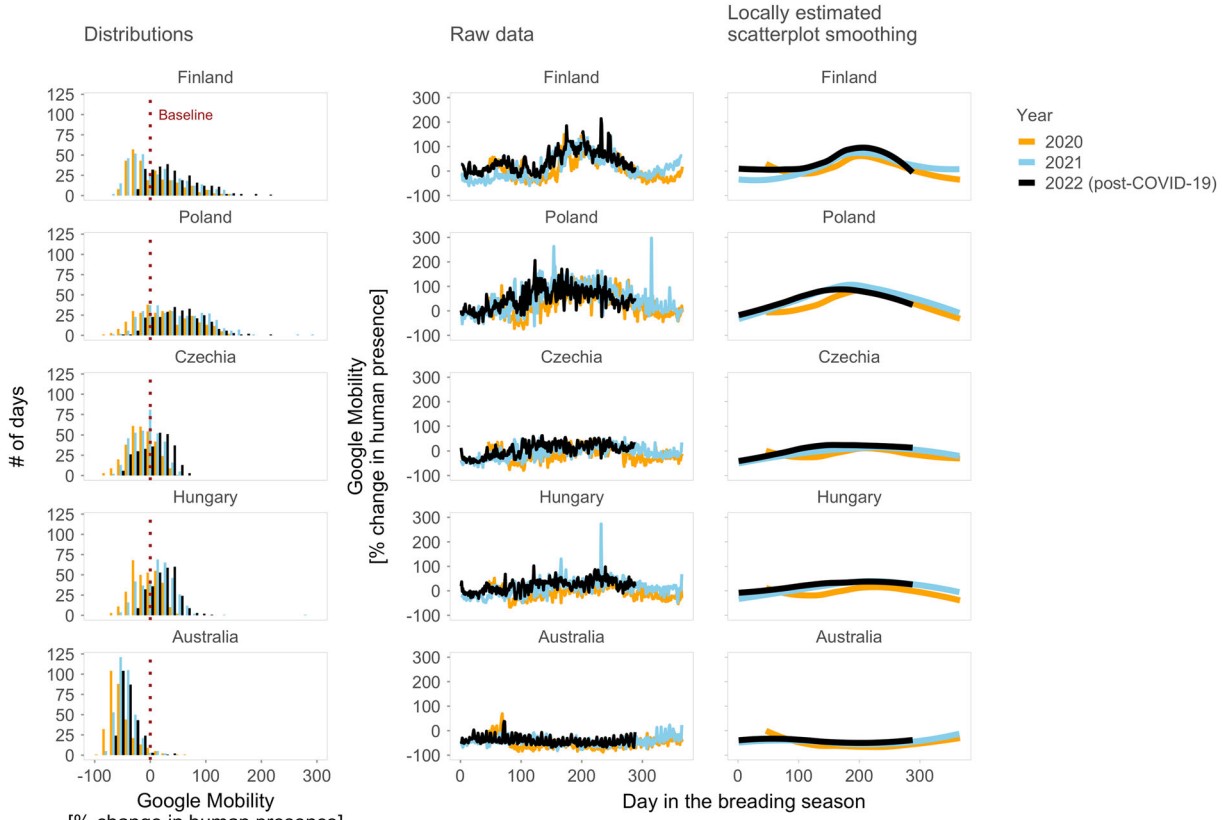

**Fig. 8 | Changes in human levels in parks within and between years and countries.** **Distributions** represent histograms of daily human levels (Google Mobility), **Raw data** the daily values and **Locally estimated scatterplot smoothing**, the smoothing of the daily data across years. Dotted vertical line in **Distributions** indicates baseline value of human levels, separating negative values that represent decreased human presence and positive values that indicate increased human presence when compared with the country- and weekday-specific baseline human levels estimated as the median value from 3 January – 6 February 2020 (see also Methods; for weekday-specific patterns see Fig. S7[43]). In Day in the breeding season, zeros represent the beginning of the breeding season. Note that Google Mobility data were unavailable for the years before the COVID-19 pandemic (i.e. before 2020) but the year 2022 was without shutdowns in the studied countries. For R-code generating the figure see interactive supporting material[43].

variable[49]. Our results might also indicate that the room for current (ongoing) selection is limited even when human presence changes on relatively long temporal scale such as in the case of the COVID-19 shutdowns or, alternatively, that the shutdowns induced changes in human presence were too weak or too short for long term adjustments in avian escape behavior. However, some studies reported changes in space use by wildlife[18,20,21], and these could arise, as our results indicate, from fixed and non-plastic animal responses to humans who changed their activities.

(ii) The urban environments might act as a filter for bold individuals[52,53,57]. Thus, the lack of consistent changes in the escape behavior of urban birds may indicate an absence (or low influx) of generally shy, less tolerant individuals and species from rural or less disturbed areas into the cities. In other words, the urban environment has filtered birds based on their inherent levels of tolerance in a similar way regardless of variation in the level of human presence. Indeed, based on genetic studies, in some species, rural populations are a source of individuals to urban populations[58,59].

(iii) Urban birds might have been already habituated to or become tolerant of variation in the levels of human presence, irrespective of the potential changes in human activity patterns[60-62]. Urban animals generally habituate to human disturbance quickly and once habituated may show only a small variation in their tolerance[48,63]. Indeed, we found that even corvids (Figs. 2–5 and S2–S5[43]), generally considered to have high levels of behavioral plasticity, do not deviate from trends observed in other species. Moreover, tolerance of individually-marked dark-eyed juncos changed little during the COVID-19 shutdowns, although individuals became more tolerant (shortened their escape distance) after shutdowns when the university campus study site was repopulated with humans[22]. Such findings are puzzling, but

indicate that habituation-like processes[64] might have already reduced plasticity. That said, there were between-site differences in the escape distance of the same species (Figs. 2–5 and S2–S5[43]) and we thus need further studies of marked individuals to better understand this phenomenon.

In sum, our findings highlight that urban birds might be most sensitive to immediate variation in human activity when making escape decisions. Whether there are generally more people during a week or year either does not seem especially influential, or alternatively such variation in human levels (given the hourly and daily variation) is insufficient to trigger any further response (beyond the immediate one). Human population and disturbance are predicted to further increase during the 21st century[65]. Therefore, animals in the Anthropocene will be increasingly forced to occupy human-altered environments, and altered environments will host more humans. Our results indicate that urban birds may not flexibly and quickly change their escape behavior to temporal variation in human presence in cities. It is unclear whether such tolerance reflects a natural selection on urban tolerance, differential settlement of individual birds in cities, or habituation-like processes that reduce plasticity. We need comprehensive genetic investigations of current populations along urban–rural/wilderness gradients combined with detailed investigation of individually-marked birds (*sensu*[52,53]) and other taxa (*sensu*[50,66]) that are repeatedly tested over time. To properly disentangle the role of differential settlement and habituation, marked birds should be investigated under experimentally manipulated human densities or when human densities change such as those that follow natural or human-induced disasters, predictable seasonal events, or rapid urban development. Our study further highlights that we still know little about how to measure human outdoor presence and which

of its features are most relevant for behavioral adjustments in free-living (urban and non-urban) animals. Answering these questions is essential if we are to successfully manage biodiversity in the Anthropocene.

## Materials and Methods

### Study areas

Flight initiation distances were collected in Finland (J.J. and field assistant), Poland (P.T.), Czechia (P.M., F.M., Y.B., K.F., A.S. and F.A.Z.), Hungary (G.M. and S.S.), and Australia (M.W. and field assistants). In the Czechia, P.M. sampled 46 sites out of 56. In Australia, M.W. sampled 32 sites out of 37. Each site was sampled by the same observer both before and during the shutdowns. All escape distances were collected during the breeding season (Europe: 1 April – 1 August; Australia: 15 August – 15 March) when most birds are territorial and active in the areas around their nests, and only in urban areas, i.e. areas with continuous urban elements, including multi-story buildings, family houses, or roads, with built-up area >50%, building density >10 buildings/ha, residential human density >10 buildings/ha[67]. Most data were collected in urban green areas, particularly parks and cemeteries. Finnish data were collected in Rovaniemi (66.500°N, 25.733°E; 64,000 inhabitants, 75–203 m a. s. l.), Polish in Poznań (52.406°N, 16.925°E; 0.53 million inhabitants, 60–154 m a. s. l.), Czech in Prague (50.083°N, 14.417°E; 1.3 million inhabitants, 177–399 m a. s. l.), Hungarian in Budapest (47.498°N, 19.041°E; 1.8 million inhabitants, 96–527 m a. s. l.), and Australian in Melbourne (37.821°S, 144.961°E; 5.2 million inhabitants, 5–169 m a. s. l.). For each city, we collected data for two breeding seasons before the pandemic, covering the 2018 and 2019 seasons immediately preceding the emergence of the COVID-19 (Finland, Hungary, Poland; and until March 2020 in Australia) and for up to two breeding seasons during the COVID-19 shutdowns (seasons starting in 2020 and 2021; for the Czech Republic, only starting in 2021). The 2019 data were unavailable for the Czech Republic, and thus we used data from 2014 and 2018. The fieldwork protocols complied with the current laws of the countries. This kind of research requires no special permits in Europe. In Australia, Animal ethics approvals (Deakin University Animal Ethics Committee Permits B10-2018 and B08-2021) and permits (DEWLP, 10008731 and 10010123) were obtained. All fieldwork was conducted following the approved guidelines. Data were collected in public places and private lands where no special permit was required. The method used to estimate avian tolerance towards human disturbance was designed to cause only brief and minimal disturbance to birds; in cities, this disturbance typically does not differ from standard background disturbance caused by other site visitors.

### Avian tolerance towards humans

Avian tolerance towards human approach was estimated by a simple but widely used method, quantifying the flight initiation distance, which is the distance at which birds escape when approached by a human observer[26,35–37]. The flight initiation distance reflects a trade-off between the fitness-related benefits of not escaping and the costs of fleeing[29,35,68,69]. The flight initiation distance estimates are highly consistent for individuals, populations, and species tested within similar contexts[26,44,49,51–53].

All data were collected by trained observers skilled in bird identification, and using a standard procedure outlined previously[32,33,37,44,46,70]. Briefly, when a focal bird (typically an adult individual) was spotted, a single observer moved at a normal walking speed (~1 ms⁻¹) directly towards the bird (with head and gaze oriented towards this bird). When the focal bird first started to escape (i.e. hopped, walked, ran, or flew away), the distance of the observer to the bird was noted. The escape distance was measured by counting the number of steps of known approximate length and converting them to meters or using a rangefinder (with ±1 m resolution). Our previous validation showed that the between-observer variation in escape distance estimates is low[71] and that distance measured by counting steps did not differ from the distance measured by a rangefinder[31]. In 79% of the cases (n = 5032 out of 6369 observations), we approached birds located on the ground. The escape distance of birds positioned above the ground (e.g. perching on vegetation) was estimated as the Euclidean distance that equals the square-

root of the sum of the squared horizontal distance and the squared height above the ground. Previous studies typically revealed no effect of perch height on birds' flight initiation distance[72–75]. We approached only individuals that were not on or next to their nests and we focused on 'relaxed birds', which we defined as those without any initial signs of distress.

Birds often occur in flocks. In these cases, we randomly selected a single individual from a flock and measured its response. All field-workers wore outdoor clothes without any bright colors. Within each city we collected data at many sites (at the level of park, cemetery, etc.); to avoid repeated sampling of the same individuals, we did not re-sample the same location during the same breeding season. Within a sampling event at a given site, individuals of the same species were sampled only if it was obvious that they were different individuals (e.g. because of their simultaneous presence or if morphological features, e.g. sex-specific coloration or age, enabled us to distinguish between different individuals). The flight initiation distances were collected during favorable weather conditions (i.e. no rain and no strong wind). In total, we collected 6369 flight initiation distance estimates for 147 bird species representing 2693 before-shutdown estimates for 68 species and 3676 during-shutdown estimates for 135 species.

### Predictors and covariates

To investigate how hourly changes in human presence in parks influence escape distance (hourly number of humans), for a subset of escape distance observation we estimated the number of humans within a 50 m radius during each escape distance trial (n = 3504; $n_{Finland (2018–2021)}$ = 761, $n_{Poland (2018)}$ = 263, $n_{Czechia (2018, 2021)}$ = 1024, $n_{Hungary (2018–2021)}$ = 1456, $n_{Australia}$ = 0; median = 1 person, mean = 3, range = 0–70).

To explore how daily and weekly changes in the levels of human presence influence escape distance, we extracted two variables. First, for each city, we extracted Google Mobility for parks, covering 2020–2022 (https://www.google.com/covid19/mobility). Google Mobility reports the change in human presence for each day of the week relative to its baseline value, estimated as the median value from 3 January – 6 February 2020. Note that Google Mobility provides the relative weekday change from the baseline (median for our data = 3%, mean = 1.5%, range = -82%–128%) and thus lacks the absolute values of human mobility; this index also lacks freely available data before 2020. Nevertheless, the year 2022 was without governmental restrictions and hence might approximate the normal (before COVID-19) human levels outdoors, albeit with a caveat as the pandemic has changed human working habits, e.g. more people currently work from home[76,77]. Second, for each country and day, we extracted data on the strength of governmental anti-pandemic measures characterized by Stringency index (https://ourworldindata.org/covid-stringency-index[42]). This index is rescaled to values from 0 to 100 (0 = no restrictions; 100 = strictest restrictions; median for our data = 42, mean = 33, range = 0–83) and represents a composite measure based on nine response indicators, including school closures, workplace closures, cancellation of public events, restrictions on public gatherings, closures of public transport, stay-at-home requirements, public information campaigns, restrictions on internal movements, and international travel controls. Note that Stringency index was not habitat-specific and did not focus directly on parks and cemeteries – the source of most of our data.

Finally, to explore how yearly changes in the levels of human presence (induced by the COVID-19 shutdowns) influenced escape distance, each observation was scored as collected before (0) or during (1) the COVID-19 shutdowns (a variable called Period).

Life-history, social, contextual, and environmental factors may influence escape responses of birds[26,35,37,44,78–81] and potentially confound associations between avian escape responses and changes in human outdoor activity. Hence, we extracted information on seven parameters. (1) The 'starting distance' was estimated as the distance to the bird (in meters) when an observer started the escape distance trial. (2) The 'flock size' was calculated as the number of all conspecific individuals moving, feeding, or perching together that were visually separated from other conspecific or mixed-species individuals. Note that we avoided approaching mixed-

species bird groups. (3) The species-specific 'body size' was approximated as body mass (in grams) and obtained as the mean of female and male values from EltonTraits 1.0 database[82]. (4) The ambient 'temperature' was estimated as the air temperature (°C) at the site during data collection either by the thermometer directly during the fieldwork (Finland) or retrospectively using online tools (Poland, Czechia, Hungary, Australia). (5) The 'time' of data collection was rounded to the nearest hour. (6) The date of data collection was noted as a 'day' since the start of the breeding season (Europe: Day 1 = 1 April; Australia: Day 1 = 15 August). (7) The 'site' represents a unique identifier of each sampled park, cemetery, city district, etc.

## Statistics and reproducibility

**Bird tolerance in relation to levels of human presence outdoors.** We explored whether changes in human levels from hour-to-hour (i.e. the actual Number of humans during escape distance trials), day-to-day (proxied by Google Mobility), week-to-week (proxied by Stringency index), or year-to-year (proxied by Period) correlated with avian escape distance. Using mixed effect models, we first fitted ln-transformed escape distance as a response and the Number of humans, Google Mobility, Stringency index, or Period as a predictor of interest, while controlling for starting distance of the observer (ln-transformed), flock size (ln-transformed), body mass (ln-transformed), temperature (also a proxy for a day within the breeding season: $r_{Pearson} = 0.48$; Fig. S6[43]), time of day and year (in case of Period fitted as a random intercept). To account for circular properties of time, time was transformed into radians ($2 \times$ time $\times \pi/24$) and fitted as sine and cosine of radians[83]. Multicollinearity among explanatory variables was checked by the correlation matrix, which suggested that correlations between variables entered simultaneously in the same model were generally weak (Fig. S6[43]). To account for the non-independence of data points[84,85], we further fitted random intercepts of weekday (day of the week), genus, species, species at a given day, country, site, and species within a site, while fitting the predictor of interest (Number of humans, Google Mobility, Stringency index or Period) as a random slope within country (i.e. allowing for different effect in each country) and in case of Period models also within site (i.e. allowing for different Period effect at each site). Fitting Number of humans, Google Mobility, Stringency index or Period as random slope at other random intercepts, or simplifying the random effects structures, produces similar results (Fig. S1, Table S1 and S2[43]). To further investigate the robustness of the findings, also because the number of sampled species (135 vs 68) and unique sampling days (141 vs 161) differed between period before and during the COVID-19 shutdowns, apart from using all observation, we fitted the models to more conservative datasets. In the case of Number of humans, Google Mobility, and Stringency index, we used a dataset with at least five observations per species ($n_{Number of humans}$ = 3446, $n_{Google Mobility}$ = 3545, $n_{Stringency index}$ = 3573), and another one with at least 10 observations per species ($n_{Number of humans}$ = 3408, $n_{Google Mobility}$ = 3399, $n_{Stringency index}$ = 3446). In the case of Period, we used a dataset with at least five observations per species and Period (i.e. at least five observations before and five during the COVID-19 shutdowns; n = 5260), the other with at least ten observations per species and each Period (n = 5106). Such alternative approaches gave nearly identical results (Fig. S1, Table S2[43]).

Although, the mixed effect models account for possible country-specific effects and are more conservative than country-specific models, we also fitted country-specific mixed-models. The country-specific models had a similar structure to the global model, excluding the random slopes and also random intercept of country and in the case of Poland also a random intercept of site because specific sites were not noted in Poland. We then verified the results from the full mixed effect model by using the country-specific estimates, standard deviations and sample sizes to estimate a meta-analytical mean using the meta.summaries function from the rmeta R-package[86].

Moreover, because of ongoing debate about whether to control escape distance for starting distance[87], we also fitted an alternative set of models without starting distance of the observer.

**Exploring human presence.** We further checked, using a limited dataset, whether actual human numbers during escape distance trials reflected Google Mobility and Stringency index. To do so we fitted mixed effect models with the Number of humans as a response variable (on original scale or on ln-scale after adding 0.01 to Number of humans), and Google Mobility or Stringency index as a predictor while also controlling for the year and the non-independence of data points by including weekday within the year as a random intercept, and Google Mobility or Stringency index as a random slope.

We further investigated the extent of day-to-day variation in Google Mobility, as well as whether Google Mobility reflected the stringency of governmental restrictions. We fitted a mixed effect model with Google Mobility as a response variable, and Stringency index as a predictor while also controlling for the year and the non-independence of data points by including weekday within the year as a random intercept and Stringency index as a random slope.

Finally, to estimate whether shutdowns changed human mobility in sampled cities, we visually compared Google Mobility in parks during shutdowns (2020–2021) and after shutdowns (2022; Figs. 7–8 and S7[43]).

**General statistical procedure.** We used R version 4.3.0 for all statistical analyses. All mixed models were fitted with the lme4 v. 1.1-29 package[88]. We then used the sim function from the arm v. 1.12-2 package and a non-informative prior distribution[89,90] to create a sample of 5000 simulated values for each model parameter (i.e. posterior distribution). We report effect sizes and model predictions by the medians, and the uncertainty of the estimates and predictions by the Bayesian 95% credible intervals represented by the 2.5 and 97.5 percentiles (i.e. 95% CI) from the posterior distribution of 5000 simulated or predicted values. We graphically inspected the goodness of fit, and the distribution of the residuals (see plots of model assumptions in[43]).

All continuous variables were standardized by subtracting the mean and dividing by the standard deviation.

Since the need for phylogenetic control depends on the phylogenetic signal in the model's residuals[91], we tested whether the residual variance contained a phylogenetic signal. To do this, we extracted the residuals from the models on the full datasets (Fig. 1; models 1a from Tables S1a–d[43]), and fitted the residuals as a new response variable in an intercept-only Bayesian linear regression fitted with STAN[92] using the brm function from the brms v. 2.17 package[93,94], with species and their phylogenetic relatedness as random effects. The phylogenetic relatedness was included as a phylogenetic covariance matrix calculated with the inverseA function in the MCMCglmm v. 2.33 package[95] from the maximum credibility tree built using maxCladeCred function in the phangorn v. 2.8.1 package[96] and 100 randomly sampled species-level phylogenetic trees (Hackett backbone) from the BirdTree online tool (http://birdtree.org)[97]. Priors were specified using the get_prior function from brms, which uses the Student's t distribution for the intercept and the standard deviation[93]. The target average proposal acceptance probability was increased to 0.99 to make the sampling more conservative to posterior distributions with high curvature[93]. Five MCMC chains ran for 5000 iterations each while discarding the first 2500 iterations as burn-in, and sampling every 5th iteration, which resulted in 2500 samples of model parameters. The independence of samples in the Markov chain was assessed using graphical diagnostics, and the convergence was estimated using the Gelman-Rubin diagnostics which was 1 for all parameters, indicating model convergence[98]. Phylogeny explained zero or little variance in the residuals from a model on the Number of humans (95% CI: 0–0.6% of variance in random effects), Google Mobility (0–0.4%), Stringency index (0–0.4%), or Period (0–0.2%) and the model without phylogeny fitted the data on residuals better than the model with phylogeny (i.e. the estimated Bayes factor in favor of non-phylogenetic model was 138 for the Number of humans, 96 for Google Mobility, 115 for Stringency index, 99 for Period; their posterior probabilities were 0.99 for all; Table S4[43]), which justifies our use of non-phylogenetic comparative methods.

All results are reproducible with the open-access data and code available from[43], which also provides visual representations of model assumptions. Visualizations were generated by the ggplot2 v. 3.3.6 package[99] and other supporting packages (for details, see[43]).

## Data availability
Data, as well as all supporting material are freely available from https://martinbulla.github.io/avian_FID_covid/[43].

## Code availability
Computer code used to generate the results of the manuscript is freely available from https://martinbulla.github.io/avian_FID_covid/[43].

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

## Acknowledgements
We thank Wolfgang Forstmeier and Lotte Schlicht for an advice on statistical analyses. M.A.W. thanks Eliza Anderson, Mickey Balzereit, Sammi Cunningham, Alisha Dabonde, Chloe Daws, Megan Dennis, Zoe Kellett, Clement Masse, Aaron Moore, Bailey Raats, Max Radvan, Rebecca Frost, Anthony Rendall, Natalie Searle, Mel Sheedy, Will Standish, Mia Stott, and Millie Toomey for help with the collection of escape responses of birds in Melbourne. Clement Masse helped us with data collection in Rovaniemi. J.J. was partly supported by the project "Responsible Tourism Planning" (2019–2023; project number 326348) funded by the Academy of Finland

(PROFI 5 Competitive funding to strengthen universities' research profiles). P.M. and P.T. were funded by the TUM – Institute for Advanced Study – Hans Fisher Senior Fellowship (to P.T.), P.M. and M. B. by the Research Excellence in Environmental Sciences (REES 003 to MB) from the Faculty of Environmental Sciences at CZU Prague. P.M. is also thankful to Anna Milosavljevičová for support during the research, M.B. to Barbora and Majlen for their patience and support.

## Author contributions
P.M. and P.T. conceived the idea. P.M., M.B., D.T.B, and M.A.W. conceptualised the study. Y.B., K.F., J.J., M.-L.K.-J., G.M., P.M., F.M., A.S., S.S, P.T., M.A.W., and F.A.Z. collected the data. M.B. with help of P.M. analyzed the data, generated the figures and prepared supplementary material. P.M. wrote the first draft of the paper as well as its revisions with help of M.B., D.T.B., M.A.W., T.A. and P.T., and all other authors (Y.B., K.F., J.J., M.-L.K.-J., G.M., F.M., A.P.M., A.S., S.S, and F.A.Z) reviewed the manuscript and its revisions.

## Funding

## Competing interests
The authors declare no competing interests.
