## [Peer Review File · Communications Biology]

Reviewers' comments:

Reviewer #1 (Remarks to the Author):

The authors undertook an impressive study, measuring fear response of urban birds across five countries, considering multiple sites, over years before and during the COVID-19 shutdowns on a total of 135 species. They considered multiple temporal scales of human activity that may affect fear response to humans, a behavior that necessarily is often different in urban environments and helps us understand their landscape of fear. While their research was limited to parks and cemeteries-areas that were generally relatively more frequented during the COVID-19 pandemic in moderate levels of governmental restrictions-and unmarked individuals, the overarching patterns that they find are relevant and interesting to the field-at-large. They find that overall, escape behavior was largely unaffected by large-scale shifts in human activity-with the exception of hourly activity, which weakly affected escape behavior. There was interesting variation in the relationships between human activity and escape behavior across sites, cities, and species. This work contributes to somewhat conflicting knowledge on the impact of human behavior (including the longer temporal scale behavioral changes caused by COVID-19) on animal behavior. It also provides evidence for the lack of plasticity in escape behavior in urban birds.

I think the results of this paper are relevant, surprising, and well-tested as a broad understanding of urban bird behavioral response to human activity. The scalar analysis and large samples of pre-COVID-19 data are particularly interesting and novel in understanding how human activity shapes animal behavior. The statistical analyses are robust and thorough, conclusions are strongly supported based on the scope and approaches of their analyses. The narrative of the article I think overemphasizes relationships between human activity levels while leaving results that are more applicable to the focus of the research in the supplement – specifically, figures on the species- and site-specific responses to human activity. I would recommend integrating these results and at least figure S3 and S10 that represent underlying complexity to the overarching conclusions. The combination of this fine-scale data and the overarching conclusions are relevant to the field and could push future research questions on behavior in urban (and non-urban) animals in the Anthropocene. In addition, added discussion on the potential limits of parks and cemeteries to answer their question on large temporal scales would be beneficial, as parks were noted for sometimes having higher human activity during parts of the COVID-19 pandemic than before when restrictions did not include park closure (e.g., Volenec et al. 2021; Csomós et al. 2023 in addition to the citations made by authors). Minor comments follow.

1. L40-41: Tracked changes can be accepted.
2. L64: FIDs were also taken after the COVID-19 shutdowns, right? It would be good to mention that in the abstract or keep language consistent and refer to COVID-19 restrictions in 2020-2021. That you have such a large timescale across countries makes this study particularly interesting and novel.
3. L91-93: An additional sentence explaining why this is important beyond understanding the impacts of COVID-19 would be beneficial here. You delve into this in your discussion, but I suggest briefly framing the importance of the study to the field at large here.
4. L96-97: Please mention how early your data collection started as this could also be interpreted as all from 2020. This is clarified later, but it would help the reader narratively and provide a better conceptualization of the study before going into results.
5. L104-113: Only the first temporal scale is numbered here. As this is a lengthy sentence, adding numbers to each measure would be helpful for the reader.
6. L133: For readers unfamiliar with flight initiation distance and the associated literature, perhaps clarify that starting distance is a known correlate of FID across multiple studies.
7. L144 "inflexible": Based on your findings, it appears that there was variation across species or population across human presence while overwhelmingly responses of urban birds did not strongly vary with respect to human presence. I would suggest toning down the word inflexible because of the (to me very interesting) results across populations, suggesting that there is flexibility in some species but

that flexibility does not have an overarching pattern.

8. L235-236 "the room for behavioral adjustments may be limited for urban birds": as far as I know and as you mention in L136-141, parks were areas that often did not experience the same severe drop in human activity as other areas in the urban environment. While your data do show shifts from baseline in human activity, and that relationships between number of humans, human activity, and stringency index vary between countries (Fig. 2), I think it's conceivable that at the sites chosen, there was not a strong enough difference in human activity for long enough to see changes in behavior that might have been recorded in areas more depopulated by humans.

9. L239: Please clarify that you found "birds did not seem to react to the change in human levels on longer temporal scales" across species, as you found interesting variation at the species level.

10. L246-247: I think that the "(ii)" is misplaced and should follow "and/or"

11. L325-328: This is a good and strong conclusion!

12. L393-394: Please include descriptive statistics of human activity as well (e.g., mean and SD). In the raw data, I noticed that there were some measures of human activity that were not integers. Was that a misread? If not, how is that possible? More details on how this is calculated would be good to include.

I hope this is of help!

References:

Csomós, G., Borza, E.M. & Farkas, J.Z. (2023) Exploring park visitation trends during the Covid-19 pandemic in Hungary by using mobile device location data. *Sci Rep* 13, 11078.
<https://doi.org/10.1038/s41598-023-38287-3>

Volenec, Z.M., Abraham, J.O., Becker, A.D., & Dobson, A.P. (2021) Public parks and the pandemic: How park usage has been affected by COVID-19 policies. *PLOS ONE* 16(5): e0251799.
<https://doi.org/10.1371/journal.pone.0251799>

Reviewer #2 (Remarks to the Author):

Summary

The authors tested how birds' flight initiation distance varied across countries before and during COVID-19 lockdowns. They found that the number of humans in the location or in the nearby area did not have a notable effect on the flight initiation. Studies investigating the temporal and spatial scale dependency of the processes are highly necessary for understanding at which scales human pressures can influence biodiversity. The manuscript is well-written and easy to follow. I have a few general comments mainly relating to the discussion points and some line-specific suggestions.

General comments

As lockdowns had different consequences in different countries (increased/decreased human activity in parks), I wonder if the hypothesis of negative relationship between flight initiation distance and human numbers is only valid for urban areas that people were not able to use during lockdowns. That is, parks were perhaps the places where people did spend time even more than normally and the difference in fear responses could be stronger in areas where the human numbers differed most drastically before and during lockdowns (such as urban non-green spaces). This comment is not meant for changing the analyses, rather for something to perhaps emphasize in the discussion.

I am wondering if you could elaborate on the baseline of the flight initiation distance for the included bird species in non-urban settings? That is, the gradient in this study is fully within urban areas so it is difficult to assess the full within-species variation in the fear responses. This suggestion is not for changing the analyses but rather for linking the study to earlier studies and extending the discussion.

I believe the biggest weakness in the analyses is the lack of Google Mobility data before lockdowns. Similarly, I only learnt from Figure 2 that human number data were missing from lockdown period from two of the five countries. I think these data gaps need to be described in a more transparent way in the methods and their effects on the obtained results discussed thoroughly.

Line-specific comments

Line 68: The sentence is unclear, could you please rephrase it. More specifically, I don't understand the comma that splits the sentence.

Line 104: What does the "(1)" refer to? If the initial idea was to have a numbered list, I agree with it. Currently, it is a little difficult to follow the long sentence with a lot of parentheses and numbering of the variables could help this issue.

Line 130: No need for the "hereafter..." as this has already been mentioned in the introduction.

Line 163: What does "human levels" refer to here? You haven't used this term elsewhere in the text so far except for the title so I suggest to define "human levels" in the introduction

Figure 1: The caption is very long and I wonder if some of the caption text could instead be included in the results text?

Figure 4: I understand that the baselines (referring to Day 0 I assume) are country-specific, but some explanation of what generally counts as Day 0 would be useful information of the figure caption.

Line 248: I think the threshold explanation is a valuable and ecologically relevant point for the observed results. I also like your suggestion for the future experiments that could test this threshold hypothesis!

Line 353: Add comma before the first "which"

Reviewers' comments:

Reviewer #1 (Remarks to the Author):

The authors undertook an impressive study, measuring fear response of urban birds across five countries, considering multiple sites, over years before and during the COVID-19 shutdowns on a total of 135 species. They considered multiple temporal scales of human activity that may affect fear response to humans, a behavior that necessarily is often different in urban environments and helps us understand their landscape of fear. While their research was limited to parks and cemeteries-areas that were generally relatively more frequented during the COVID-19 pandemic in moderate levels of governmental restrictions-and unmarked individuals, the overarching patterns that they find are relevant and interesting to the field-at-large. They find that overall, escape behavior was largely unaffected by large-scale shifts in human activity-with the exception of hourly activity, which weakly affected escape behavior. There was interesting variation in the relationships between human activity and escape behavior across sites, cities, and species. This work contributes to somewhat conflicting knowledge on the impact of human behavior (including the longer temporal scale behavioral changes caused by COVID-19) on animal behavior. It also provides evidence for the lack of plasticity in escape behavior in urban birds.

Many thanks for your positive appraisal of our manuscript, as well as for your constructive comments. We very much appreciate that.

I think the results of this paper are relevant, surprising, and well-tested as a broad understanding of urban bird behavioral response to human activity. The scalar analysis and large samples of pre-COVID-19 data are particularly interesting and novel in understanding how human activity shapes animal behavior. The statistical analyses are robust and thorough, conclusions are strongly supported based on the scope and approaches of their analyses.

We find it encouraging that you appreciate the large data sample, multiple temporal scales, and find the analysis robust and the analyses from within the Supplement of interest.

The narrative of the article I think overemphasizes relationships between human activity levels while leaving results that are more applicable to the focus of the research in the supplement – specifically, figures on the species- and site-specific responses to human activity. I would recommend integrating these results and at least figure S3 and S10 that represent underlying complexity to the overarching conclusions. The combination of this fine-scale data and the overarching conclusions are relevant to the field and could push future research questions on behavior in urban (and non-urban) animals in the Anthropocene.

We greatly appreciate your comment and encouragement to bring finer scale results into the main text. We thus moved figure S7-S10 (as Fig. 2-5) into the main text. We have thus rearranged the order of the figures, both in the main text and supplementary material (see table below).

Old order	New order
Figure 1	Figure 1
Figure S7	Figure 2
Figure S8	Figure 3
Figure S9	Figure 4
Figure S10	Figure 5
Figure 2	Figure 6
Figure 3	Figure 7
Figure 4	Figure 8
Figure S1	Figure S1
Figure S3	Figure S2
Figure S4	Figure S3
Figure S5	Figure S4
Figure S6	Figure S5
Figure S2	Figure S6
Figure S11	Figure S7

If the Editor wishes to move any of the original main text figures into the SI or to bring more SI figures into the main text, we are happy to do so.

In addition, added discussion on the potential limits of parks and cemeteries to answer their question on large temporal scales would be beneficial, as parks were noted for sometimes having higher human activity during parts of the COVID-19 pandemic than before when restrictions did not include park closure (e.g., Volenec et al. 2021; Csomós et al. 2023 in addition to the citations made by authors).

We also welcome your call for broadening the discussion. We discuss the limitations of focusing on parks and cemeteries (L203-208, L240-242).

Minor comments follow.

1. L40-41: Tracked changes can be accepted.

Changed.

2. L64: FIDs were also taken after the COVID-19 shutdowns, right? It would be good to mention that in the abstract or keep language consistent and refer to COVID-19 restrictions in 2020-2021. That you have such a large timescale across countries makes this study particularly interesting and novel.

We regret being unclear. We have collected FIDs only before pandemic (i.e. before 2020) and during the pandemic (2020-2021). We thus rewrote this part of an Abstract (L59-63).

3. L91-93: An additional sentence explaining why this is important beyond understanding the impacts of COVID-19 would be beneficial here. You delve into this

in your discussion, but I suggest briefly framing the importance of the study to the field at large here.

Thank you – we expanded this part (L96-99).

4. L96-97: Please mention how early your data collection started as this could also be interpreted as all from 2020. This is clarified later, but it would help the reader narratively and provide a better conceptualization of the study before going into results.

We added “in years 2014, 2018, 2019” (L104).

5. L104-113: Only the first temporal scale is numbered here. As this is a lengthy sentence, adding numbers to each measure would be helpful for the reader.

Thank you, we added the numbers here.

6. L133: For readers unfamiliar with flight initiation distance and the associated literature, perhaps clarify that starting distance is a known correlate of FID across multiple studies.

We added this clarification (L140).

7. L144 “inflexible”: Based on your findings, it appears that there was variation across species or population across human presence while overarching responses of urban birds did not strongly vary with respect to human presence. I would suggest toning down the word inflexible because of the (to me very interesting) results across populations, suggesting that there is flexibility in some species but that flexibility does not have an overarching pattern.

Thank you for this comment. We delete the whole sentence because we found it repetitive.

8. L235-236 “the room for behavioral adjustments may be limited for urban birds”: as far as I know and as you mention in L136-141, parks were areas that often did not experience the same severe drop in human activity as other areas in the urban environment. While your data do show shifts from baseline in human activity, and that relationships between number of humans, human activity, and stringency index vary between countries (Fig. 2), I think it’s conceivable that at the sites chosen, there was not a strong enough difference in human activity for long enough to see changes in behavior that might have been recorded in areas more depopulated by humans.

We agree that parks may have not experienced as severe drops in human activity as other urban areas and hence expanded the discussion: “Whether similar shifts in human levels would induce changes, for examples, in escape distance of rural birds that are unhabituated to humans and tend to have longer escape distances ^{26,28}, awaits testing. Moreover, other urban areas than parks might have experienced stronger changes in human levels (perhaps reaching a required threshold), but whether birds changed their escape response there is also unknown.” (L203-208).

Note however, that parks often experienced substantial increase in human activity during shutdowns (e.g. Zhao et al. 2023, *Sci Rep* 13:1–11; Geng et al. 2021, *J For Res* 32:553–567; Volenec et al. 2021, *PLoS One* 16:e0251799; Csomós et al. 2023 *Sci Rep* 13:1–12).

Importantly, we used COVID-19 shutdowns as a means that increased variation in human levels. Such data together with pre-COVID-19 data provide highly variable human levels (now Fig. 6 and descriptive data in the Methods L355-356, 362-363, 370-371), at least on shorter temporal scales. We agree (and discuss) that for long term bird adjustments the shutdowns may have generated too low or too short of a change in human levels, e.g. L203-208, L240-242.

9. L239: Please clarify that you found “birds did not seem to react to the change in human levels on longer temporal scales” across species, as you found interesting variation at the species level.

We added “across-species” (L189).

10. L246-247: I think that the “(ii)” is misplaced and should follow “and/or”

Changed.

11. L325-328: This is a good and strong conclusion!

Thank you.

12. L393-394: Please include descriptive statistics of human activity as well (e.g., mean and SD). In the raw data, I noticed that there were some measures of human activity that were not integers. Was that a misread? If not, how is that possible? More details on how this is calculated would be good to include.

All hourly measures of human levels in our data are integers and so are Google Mobility measures. Stringency index represents a composite measure based on nine response indicators and hence is not an integer. We now included descriptive statistics for these three measures of human activity (L355-356, 362-363, 370-371).

I hope this is of help!

Yes, we find your constructive comments extremely helpful and appreciate your investment.

References:

Csomós, G., Borza, E.M. & Farkas, J.Z. (2023) Exploring park visitation trends during the Covid-19 pandemic in Hungary by using mobile device location data. *Sci Rep* 13, 11078. <https://doi.org/10.1038/s41598-023-38287-3>

Volenec, Z.M., Abraham, J.O., Becker, A.D., & Dobson, A.P. (2021) Public parks and the pandemic: How park usage has been affected by COVID-19 policies. *PLOS ONE* 16(5): e0251799. <https://doi.org/10.1371/journal.pone.0251799>

Thanks – now cited in the discussion.

Reviewer #2 (Remarks to the Author):

Summary

The authors tested how birds' flight initiation distance varied across countries before and during COVID-19 lockdowns. They found that the number of humans in the location or in the nearby area did not have a notable effect on the flight initiation. Studies investigating the temporal and spatial scale dependency of the processes are highly necessary for understanding at which scales human pressures can influence biodiversity. The manuscript is well-written and easy to follow. I have a few general comments mainly relating to the discussion points and some line-specific suggestions.

Many thanks for your comments, we very much appreciate them.

General comments

As lockdowns had different consequences in different countries (increased/decreased human activity in parks), I wonder if the hypothesis of negative relationship between flight initiation distance and human numbers is only valid for urban areas that people were not able to use during lockdowns. That is, parks were perhaps the places where people did spend time even more than normally and the difference in fear responses could be stronger in areas where the human numbers differed most drastically before and during lockdowns (such as urban non-green spaces). This comment is not meant for changing the analyses, rather for something to perhaps emphasize in the discussion.

Thank you for this comment. Indeed, lockdowns had different effects on human levels in different countries (Fig. 6-8 and S7). Importantly, the general expectation of negative relationship between human levels and escape distance should hold across contexts, i.e. regardless of whether human levels increased or decreased. We agree that bird responses might have been stronger in places where the change in human levels were the greatest (see added discussion L199-208) and we indeed discuss that changes in parks might not have been severe or long enough (L203-208, L240-242).

I am wondering if you could elaborate on the baseline of the flight initiation distance for the included bird species in non-urban settings? That is, the gradient in this study is fully within urban areas so it is difficult to assess the full within-species variation in the fear responses. This suggestion is not for changing the analyses but rather for linking the study to earlier studies and extending the discussion.

Many thanks for this comment. In general, rural populations of a given species are less tolerant of humans (i.e. have longer escape distances). We expanded our discussion (L203-208).

I believe the biggest weakness in the analyses is the lack of Google Mobility data before lockdowns. Similarly, I only learnt from Figure 2 that human number data were missing from lockdown period from two of the five countries. I think these data gaps

need to be described in a more transparent way in the methods and their effects on the obtained results discussed thoroughly.

We agree that our study would greatly benefit from having Google Mobility data before 2020. As we note in the Methods (L364) and Fig. 8 caption, such data were however unavailable.

Note that human number data were completely missing only for Australia (Fig. 1); in a case of Poland, we missed human number data during the pandemic (hence, comparison with Google Mobility/Stringency Index was not possible) as stated in a present Fig. 6 and through sample sizes in Methods (L355-356) where we, for completeness, added also Australia.

Line-specific comments

Line 68: The sentence is unclear, could you please rephrase it. More specifically, I don't understand the comma that splits the sentence.

We rewrote this sentence.

Line 104: What does the "(1)" refer to? If the initial idea was to have a numbered list, I agree with it. Currently, it is a little difficult to follow the long sentence with a lot of parentheses and numbering of the variables could help this issue.

Thanks for spotting this mistake. We are now using a numbered list of different proxies of human levels across different temporal scales.

Line 130: No need for the "hereafter..." as this has already been mentioned in the introduction.

Deleted.

Line 163: What does "human levels" refer to here? You haven't used this term elsewhere in the text so far except for the title so I suggest to define "human levels" in the introduction

Human levels refer to number of humans, i.e. levels of human presence, and we now define this term in the Introduction (L93-94).

Figure 1: The caption is very long and I wonder if some of the caption text could instead be included in the results text?

We now shortened the Fig. 1 caption as all deleted information was (and still is) already provided in the method section.

Figure 4: I understand that the baselines (referring to Day 0 I assume) are country-specific, but some explanation of what generally counts as Day 0 would be useful information of the figure caption.

We regret being unclear. The baseline indicates absolute median number of humans from 3 January – 6 February 2020 for a given weekday and country. The Google Mobility data then show percentage change in human levels for a given weekday and country given the baseline. To clarify, we rewrote the legend (note that Fig. 4 is now Fig. 8).

Line 248: I think the threshold explanation is a valuable and ecologically relevant point for the observed results. I also like your suggestion for the future experiments that could test this threshold hypothesis!

Many thanks for this much appreciated feedback!

Line 353: Add comma before the first "which"

Added.

REVIEWERS' COMMENTS:

Reviewer #1 (Remarks to the Author):

I commend the authors on a fine job considering the comments of myself and the second reviewer. Points that needed more clarity and added information have been added appropriately. I believe this manuscript is suitable for publication. I only have minor stylistic and grammatical comments as follows, as well as one line that needs further clarity. Line numbers refer to the clean document.

Title: As you have changed the language in the manuscript from "human levels" to "human presence" (which I agree reads better), I suggest changing the language in the title, as well, to: "Urban birds' tolerance towards humans was largely unaffected by COVID-19 shutdown induced variation in human presence."

L59: "human levels" can be changed to "levels of human presence"

L126, 127, and 129: "human numbers" reads a little awkward; "number of humans" would be clearer.

L235: Add "The" to "urban environments"

L264-265: "to a temporarily changed human presence in cities" is unclear as it is contextually implied that changes in human presence is temporal; I suggest rewording. Do you mean "temporal variation in human presence" or "changes in human presence at different temporal scales"?

L273: There is a typo in the word "predictable."

Figure 7: There is a typo in the y-axis label ("Mobiligy" instead of "Mobility").

Reviewer #2 (Remarks to the Author):

The authors have done a thorough job at accounting for the feedback in their revised manuscript. I (previously Reviewer #2) am happy with the changes the authors have made to the manuscript. I only have a few small comments below. I look forward to seeing this work published!

Title: You have largely changed "human levels" to "human presence" especially in the discussion. I wonder if it would be better to do so also in the title as I still think that "human level" is not a commonly known term. In general, consistency in terminology throughout the manuscript text is advisable.

Line 206 (line number refers to the track-changes version of the manuscript): Remove the first "other" in the sentence

Figures 2-5: These figures include a massive amount of information and might be quite overwhelming for the reader. Alternative to moving them from supplementary material to the main text is to take a few example species from these figures and show the results only for them in the main text and keep the full figures in the supplementary material. As the manuscript stands now, the details in each figure are too small to see, including the species names, degree of overlap in point observation, and the bright yellow color against the white background.

Data availability: It's great that you're sharing your data openly. However, I would hope to see the

data and the code shared on a permanent repository with a DOI rather than a GitHub repository that doesn't have a similar guarantee of a permanence.

Reviewer #1 (Remarks to the Author):

I commend the authors on a fine job considering the comments of myself and the second reviewer. Points that needed more clarity and added information have been added appropriately. I believe this manuscript is suitable for publication. I only have minor stylistic and grammatical comments as follows, as well as one line that needs further clarity. Line numbers refer to the clean document.

Thank you for this encouraging words. Your constructive comments helped us in improving the manuscript!

Title: As you have changed the language in the manuscript from "human levels" to "human presence" (which I agree reads better), I suggest changing the language in the title, as well, to: "Urban birds' tolerance towards humans was largely unaffected by COVID-19 shutdown induced variation in human presence."

Done.

L59: "human levels" can be changed to "levels of human presence"

Done.

L126, 127, and 129: "human numbers" reads a little awkward; "number of humans" would be clearer.

Done.

L235: Add "The" to "urban environments"

Done.

L264-265: "to a temporarily changed human presence in cities" is unclear as it is contextually implied that changes in human presence is temporal; I suggest rewording. Do you mean "temporal variation in human presence" or "changes in human presence at different temporal scales"?

We mean the first. Many thanks, changed.

L273: There is a typo in the word "predictable."

Thanks, corrected,

Figure 7: There is a typo in the y-axis label ("Mobiligy" instead of "Mobility").

Thank you for spotting this. Done.

Reviewer #2 (Remarks to the Author):

The authors have done a thorough job at accounting for the feedback in their revised manuscript. I (previously Reviewer #2) am happy with the changes the authors have made to the manuscript. I only have a few small comments below. I look forward to seeing this work published!

We are grateful that you appreciate our affords – we very much appreciate your constructive comments!

Title: You have largely changed "human levels" to "human presence" especially in the discussion. I wonder if it would be better to do so also in the title as I still think that "human level" is not a commonly known term. In general, consistency in terminology throughout the manuscript text is advisable.

Done.

Line 206 (line number refers to the track-changes version of the manuscript): Remove the first "other" in the sentence

Done.

Figures 2-5: These figures include a massive amount of information and might be quite overwhelming for the reader. Alternative to moving them from supplementary material to the main text is to take a few example species from these figures and show the results only for them in the main text and keep the full figures in the supplementary material. As the manuscript stands now, the details in each figure are too small to see, including the species names, degree of overlap in point observation, and the bright yellow color against the white background.

The reviewer 1 in the first round of reviews found some of the supplementary figures essential to the text (e.g. because they demonstrate the variability and complexity of the matter) and called for bringing those into the main text, which we did. You seem unsure of this step. Given that you are not a priory against the first reviewer's recommendation, we chose to keep the figures within the main text, unless the Editor wishes otherwise.

Data availability: It's great that you're sharing your data openly. However, I would hope to see the data and the code shared on a permanent repository with a DOI rather than a GitHub repository that doesn't have a similar guarantee of a permanence.

The final snapshot of the GitHub repository will be accompanied by DOI via Open Science Framework or Zenodo, once the main document will be final.